# Naringin and Naringenin Polyphenols in Neurological Diseases: Understandings from a Therapeutic Viewpoint

**DOI:** 10.3390/life13010099

**Published:** 2022-12-29

**Authors:** Talha Bin Emran, Fahadul Islam, Nikhil Nath, Hriday Sutradhar, Rajib Das, Saikat Mitra, Mohammed Merae Alshahrani, Abdulaziz Hassan Alhasaniah, Rohit Sharma

**Affiliations:** 1Department of Pharmacy, BGC Trust University Bangladesh, Chittagong 4381, Bangladesh; 2Department of Pharmacy, Faculty of Allied Health Sciences, Daffodil International University, Dhaka 1207, Bangladesh; 3Department of Pharmacy, International Islamic University Chittagong, Chittagong 4318, Bangladesh; 4Department of Pharmacy, Faculty of Pharmacy, University of Dhaka, Dhaka 1000, Bangladesh; 5Department of Clinical Laboratory Sciences, Faculty of Applied Medical Sciences, Najran University, P.O. Box 1988, Najran 61441, Saudi Arabia; 6Department of Rasa Shastra and Bhaishajya Kalpana, Faculty of Ayurveda, Institute of Medical Sciences, Banaras Hindu University, Varanasi 221005, Uttar Pradesh, India

**Keywords:** naringin, naringenin, polyphenols, neurological disease, dietary interventions

## Abstract

The glycosides of two flavonoids, naringin and naringenin, are found in various citrus fruits, bergamots, tomatoes, and other fruits. These phytochemicals are associated with multiple biological functions, including neuroprotective, antioxidant, anticancer, antiviral, antibacterial, anti-inflammatory, antiadipogenic, and cardioprotective effects. The higher glutathione/oxidized glutathione ratio in 3-NP-induced rats is attributed to the ability of naringin to reduce hydroxyl radical, hydroperoxide, and nitrite. However, although progress has been made in treating these diseases, there are still global concerns about how to obtain a solution. Thus, natural compounds can provide a promising strategy for treating many neurological conditions. Possible therapeutics for neurodegenerative disorders include naringin and naringenin polyphenols. New experimental evidence shows that these polyphenols exert a wide range of pharmacological activity; particular attention was paid to neurodegenerative diseases such as Alzheimer’s and Parkinson’s diseases, as well as other neurological conditions such as anxiety, depression, schizophrenia, and chronic hyperglycemic peripheral neuropathy. Several preliminary investigations have shown promising evidence of neuroprotection. The main objective of this review was to reflect on developments in understanding the molecular mechanisms underlying the development of naringin and naringenin as potential neuroprotective medications. Furthermore, the configuration relationships between naringin and naringenin are discussed, as well as their plant sources and extraction methods.

## 1. Introduction

The term “neurological disease” is often used to refer to anything that affects the nervous system. In the brain, spinal cord, or other nerves, structural, metabolic, or electrical dysfunctions may cause a broad range of symptoms. These symptoms include altered states of consciousness, convulsions, muscular weakness, poor coordination, a loss of feeling, and a lack of sensation. Other symptoms include disorientation, pain, and discomfort. Extensive study has led to the discovery of several neurological abnormalities, some of which are somewhat common while others are relatively uncommon [1,2]. Some of these problems are inherited while others are not. Neurology and clinical neuropsychology can identify and treat these illnesses using the appropriate diagnostic and therapeutic methods. Although encased in the skull and spinal vertebrae and chemically isolated by the blood–brain barrier, the brain and spinal cord are susceptible organs [3]. Because of their placement just beneath the epidermis, nerves are nevertheless vulnerable to harm despite the protective layers of tissue around them. Individual neurons and the neural circuits and nerves they form are susceptible to damage to their electrical and structural integrity [4].

Phytochemicals may be broken down into several different classes, such as flavanols, flavan-3-ols, isoflavones, flavanones, anthocyanidins, and flavones [5]. It has been shown that flavonoids promote apoptosis and decrease metastasis, angiogenesis, and proliferation in the setting of carcinogenesis by interfering with various cell signaling pathways. The flavanone-7-O-glycoside naringin is found in many plants, while citrus fruits are the most common place to find it. A flavanone called naringenin is its main component [6,7]. Many different signaling pathways and signaling molecules are affected by this chemical. Its pharmacological properties include but are not limited to an antioxidant, an anti-inflammatory, an anti-apoptotic, an anti-tumor, and an anti-viral, as well as effects on metabolic syndrome, bone regeneration, neurological illnesses, cardiovascular disease, and genetic damage [8]. Naringin may cause drug interactions due to its ability to inhibit cytochrome P450 enzymes, including CYP3A4 and CYP1A2 [9].

Neurological disorders may be alleviated with the help of naringenin (4′,5,7-trihydroxy flavanone), a flavonoid abundant in the peels of citrus fruits (particularly grapes and tomatoes) [10]. Among the top dietary sources of this noteworthy flavonoid are grapefruits and oranges. A few of the many medical advantages of this drug include preventing or reversing weight gain, enhancing metabolic health, and restoring typical lipid profiles in patients with dyslipidemia [11]. Among naringenin’s many biological properties is an antioxidant effect. It has been shown that they may help with pain, inflammation, kidney health, and even nerve function. Research has revealed that naringenin may help reduce pain and inflammation associated with several medical disorders [12]. Mechanistically, naringenin is a pleiotropic molecule that inhibits leukocyte recruitment, pro-inflammatory cytokine release, and cytokine-induced analgesia by modulating transient receptor potential (TRP) channels and triggering the nitric oxide (NO)/cyclic guanosine monophosphate (cGMP)/protein kinase G(PKG)/adenosine triphosphate (ATP The compound’s low bioavailability and limited brain access are significant roadblocks to broader use despite naringenin’s promise in treating neurological diseases [13,14]. The therapeutic potential of naringin and naringenin for various symptoms related to neurological diseases has been shown by encouraging results from preclinical investigations [15].

The principal objective of this review was to report on the progress made toward understanding the molecular mechanisms behind the development of naringin and naringenin as potential neuroprotective therapeutics. Additionally, the configuration links between naringin and naringenin, as well as their plant sources and extraction methods, are described.

## 2. Methodology

The keywords naringin and naringenin, neurological diseases, botanical sources, and neuroprotective action were used to search the following databases: PubMed, Scopus, and Web of Science. Up to 2022, English-language research reports, reviews, and original research articles were considered and studied. An algorithm was used that followed the flowchart in Figure 1 and had all of the steps and requirements for choosing the required literature, in accordance with the recommendations of Page et al. [16].

## 3. Naringin and Naringenin 

The flavanone naringenin and the glycosylated derivative naringin are in high concentrations in grapefruit and other citrus fruits. The antioxidant and anti-inflammatory properties of flavonoids have gained widespread recognition [17]. When naringenin is added to SH-SY5Y cells or when 6-OHDA is injected into mice, the cells are protected in an Nrf2-dependent manner, just like the dopaminergic neurons [18]. This is due to the complete elimination of protective effects and expression of Nrf2-dependent cytoprotective genes after treatment with Nrf2 short interfering RNA [19]. Naringenin prevented rotenone-induced structural alterations in muscle and motor impairment in rat models when given after the administration of the medication. The expression of DJ-1 and chaperone-associated E3 ligase was increased in the striatum and SN after treatment with naringenin [20]. Target proteins are ubiquitinated by DJ-1 and a chaperone-associated E3 ligase and then sent on to be degraded by the proteasome. In a rotenone model of Parkinson’s disease, naringenin was found to have neuroprotective effects [21,22]. Naringin may aid dopaminergic neuron recovery after injury if given soon after the damage occurs. Dopaminergic neurons were preserved, GDNF levels in the SN were restored, and the number of ionized calcium-binding adaptor molecule 1 (Iba-1) and tumor necrosis factor-alpha (TNF-α) immunoreactive neurons in the striatum were reduced after pretreatment with naringin in rats with a unilateral MPP+-lesion. Eukaryotic initiation factor 4E-binding protein 1 (4E-BP1) and growth differentiation and neurotrophic factor (GDNF) were upregulated in the SN after a single injection of naringin [23].

On top of that, it is an effective antioxidant. During fasting and stimulated states, rapid glucose uptake is impaired in BC. Insulin stimulates phosphoinositide 3-kinase PIP3/Akt and mitogen-activated protein kinase (MAPK) activity, and naringenin prevents this MAPK. It decreases TNF-α and COX-2 levels and raises the transcription factor Nrf2 [24,25].

## 4. Botanical Sources

Flavonoids are phenolic compounds associated with a wide range of biological functions. There are more than 4000 different flavonoids known to science, most of which are found in their natural, unaltered plant-based forms. Flavonoids can be a dietary supplement [26,27]. Grapefruit and other citrus fruits get their distinctive bitter taste from flavonoid naringin. Although the number of flavonoids taken from food may be large and the flavonoids exhibit potential biological action, they have attracted significantly less attention than flavanols and isoflavones [28,29].

In most cases, the researchers focused on flavanols and isoflavones—intracellular cycling of naringenin, hesperidin, and its glycosylated derivatives, naringenin, hesperidin, and rutin. Grapefruit, bergamot, sour orange, tart cherry, tomato, chocolate, Greek oregano, water mint, and beans are all foods and plants that contain norepinephrine or its glycosides [29,30]. 

## 5. Neuroprotective Action 

### 5.1. Alzheimer’s Disease

Alzheimer’s disease (AD) is the most common type of dementia in the industrialized world and is a severe public health problem [31]. Several different molecular processes caused AD, but its exact pathophysiology is still poorly understood. In several pre-clinical studies, naringin and some of its derivatives, such as naringenin, changed these pathways in ways that could be used to treat AD [32,33]. The disease is caused by the death of cholinergic neurons in the frontal lobe and the formation of Amyloid-β (Aβ) plaques outside the body [34]. 

Memantine, an antagonist of the NMDA glutamate receptor, breaks down at a much slower rate than acetylcholinesterase (AChE) inhibitors, such as donepezil aricept, which is often used to treat the symptoms of AD [35,36]. Synaptic dysfunction, in which synapses are damaged, cells are killed, and mental impairments occur, results from consuming too much. This dysfunction can be fixed entirely with the proper treatment. It is essential to understand how Aβ is currently in charge of producing and storing memories and how this affects synaptic plasticity in the brain network. Evidence suggests that calcium/calmodulin-dependent protein kinase II (CaMKII) is a critical synaptic target for Aβ-induced synaptic depression [37,38]. Several plant species high in flavonoids have been used in traditional medicine for hundreds of years. Epidemiological and dietary studies on both people and animals have shown that these flavonoids protect against and slow down neurodegeneration, especially regarding the cognitive decline that accompanies aging [39]. The flavonoid glycoside naringin, found in citrus fruits in large amounts, is effective against many diseases and conditions, such as cancer, inflammation, ulcers, osteoporosis, and apoptosis. Naringin has been shown to improve behavior and thinking in animal models of epilepsy caused by kainic acid and Huntington’s disease caused by 3-nitropropionic acid [40]. The effects of colchicine and D-galactose on learning and memory are also undone by treatment with naringin. Naringenin has been shown to improve insulin signaling and cognitive ability in the brain and reduce the effects of intracerebroventricular-streptozotocin on neurodegeneration caused by AD (Table 1 and Figure 2) [41].

### 5.2. Parkinson’s Disease

Parkinson’s disease (PD) is a degenerative neurological disease that results in impaired motor function due to dopaminergic neuropathy in the substantia nigra [42,43,44]. Genetic predisposition may play a role in developing mitochondrial damage and oxidative stress; however, additional molecular routes exist. In recent years, oxidative stress has been investigated as a potential mechanism in neurodegeneration, which is only one example of many similarities between AD and PD. The antioxidant properties of flavonoids, alkaloids, and other polyphenols are coming to the fore. Plant chemicals may modulate enzymes and metabolic signaling pathways to reduce ROS production. Bioactive metabolite flavonoids have anti-oxidative actions on the liver’s metabolic pathway [45,46,47]. 

The antioxidant and neuroprotective properties of naringin have been studied. Asahina and Inubuse found the blueprints for naringin in 1928 [48,49]. Naringenin is linked to naringin through the C-7 hydroxyl group. Metabolites of naringenin may be found in phases I and II of the drug’s metabolism. Glucose is the source of naringin’s bitter flavor. It reacts with potassium hydroxide or other essential substances to form 1,3-diphenylpropan-1-one, which has a scent reminiscent of menthol [50,51,52]. It has been shown that naringin may neutralize ROS, scavenge superoxide, suppress xanthine oxide, decrease lipid peroxidation, and decrease the permeability of oxygen-stimulated K+ erythrocytes. Naringin’s antioxidant potentials may aid in treating neurology and diabetes. Degeneration of nerve cells in the striatum and substantia nigra pars compacta kills dopamine-producing brain cells, leading to Parkinson’s disease (Table 1 and Figure 3) [53]. Neurons in the substantia nigra are affected as microglia are activated, and protein clumps form. Causes of neurodegeneration include oxidative stress, dopamine depletion, and neuroinflammation. In addition to boosting dopamine, naringenin also reduces inflammation [54]. By activating Nrf2/ARE and its downstream target genes, including HO-1 and glutathione cysteine ligase regulatory subunit, naringenin protects mice against 6-hydroxydopamine-induced dopaminergic neurodegeneration and oxidative damage. By inhibiting and caspase 3, naringenin was able to prevent apoptosis. Grapefruit suppresses CYP3A4 much more so than naringin. With less naringin than grapefruit juice, orange juice also stops CYP3A4. Preclinical tests have shown that naringin is not transformed into naringenin in cultivated cells [55,56,57].
life-13-00099-t001_Table 1Table 1Preclinical findings on the use of naringin and naringenin polyphenols in neurological disorders.DiseaseCompoundDose/Conc.Study ModelFindingsReferencesAlzheimer DiseaseNaringin50, 100 and 200 mg/kg; PO) for twentyone daysICV-STZ ratsRestoration of cognitive deficits in ICV-STZ rat along with mitigation of mitochondrial dysfunction mediated oxido-nitrosative stress and cytokine release[58]Naringin50 or 100 mg/kg/dayAPPswe/PSΔE9 transgenic miceReduction in plaque burden and an increase in glucose uptake through the inhibition of GSK-3β activity[59]Naringin40 and 80 mg/kgWistar ratsProtection against ICV β-A1–42 and intranasal manganese induced memory dysfunction possibly due to its antioxidant, anti-inflammatory, anti-amyloidogenesis[60]Naringin100 mg/kg/dayMiceNeuroprotective effects through a variety of mechanisms, including amyloid β metabolism, Tau protein hyperphosphorylation, acetyl cholinergic system, glutamate receptor system, oxidative stress and cell apoptosis[61]Naringenin70–210 µg/mLPC12 cellsInhibition of AChE activity[62]Naringenin50 mg/kgMale albino Wistar ratsReduced oxidative stress markers: 4-HNE, MDA, TBARS, H_2_O_2_, PC, GSH in the hippocampus; Increase antioxidant level: GPx, GR, GST, SOD, CAT and Na+/K+-ATPase in the hippocampus[63]Naringenin25, 50 and 100 mg/kgMale Sprague-Dawley ratsIncreased the mRNA expression of INS and INSR in cerebral cortex and hippocampus. In addition, NAR reversed ICV-STZ induced Tau hyper-phosphorylation in both hippocampus and cerebral cortex through downregulation of GSK-3β activity[41]Naringenin25, 50 and 100 μM and 1.5, 3.0 and 4.5 mg/kgPC12 cells and male ICR miceDecreased ROS level and LDH activity[64]Naringenin25 and 50 mg/kgMale Sprague-Dawley ratsDecreased oxidative stress by depleting elevated lipid peroxide and nitric oxide and elevating reduced glutathione levels and exert cholinergic function through the inhibition of elevated ChE activity[65]Naringin dihydrochalcone100 mg/kgAPPswe/PS1ΔE9 (APP/PS1) transgenic miceReduction in amyloid plaque burden and Aβ levels, suppression of neuroinflammation and promotion of neurogenesis[66]Naringin
40 and 80 mg/kg, PO
Male Wistar ratsImprovement in the cognitive performance and attenuated oxidative damage, as evidenced by lowering of malondialdehyde level and nitrite concentration and restoration of superoxide dismutase, catalase, glutathione S-transferase, and reduced glutathione levels, and acetylcholinesterase activity[67]Naringin30 or 60 mg/kg/dayNMRI male miceReduction of Aβ plaque numbers in CA1, CA3, and DG areas of the hippocampus[68]Naringin25, 50 and 100 mg/kg POWestar ratsReduced lipid peroxidation, restored reduced superoxide dismutase and catalase) and acetylcholine esterase activity were significantly decreased[69]N,N′-1,10-Bis(Naringin) Triethylenetetraamine10–200 μM
PC12 cells

Deceased the level of ROS in Cu^2+^-Aβ_1-42_-treated PC12 cells and elevate the SOD activity in Cu^2+^-Aβ_1-42_-treated PC12 cells
[70]Naringin2.5, 5 and 10 mg/kgSwiss miceIncreased the activities of superoxide dismutase and catalase, and glutathione and decreased malondialdehyde and nitrite contents, and reduced brain acetylcholinesterase activity in mice brains[71]Naringin80 mg/kgWistar albino ratsImprovement of the Aβ-induced cholinergic dysfunction and increase in the activity of AChE in rat hippocampus, prefrontal cortex, and amygdala. Furthermore, naringin attenuated Aβ-induced decrease in mitochondrial function, integrity, and bioenergetics as well as mitochondrial and cytosolic calcium level in all the brain regions. Moreover, reversal of Aβ-induced increase in apoptosis and level of mitochondrial calcium uniporter and decrease in the level of hemeoxygenase-1[72]Naringenin100 mg/kg, orallymale Wistar ratsLowered hippocampal MDA content[73]Parkinson DiseaseNaringin80 mg/kgrat modelProtection ofthe nigrostriatal DA projection by increasing glialcell line-derived neurotrophic factor expression and decreasingTNF-a expression in DA neurons and microglia[74]Naringin8 or 80 mg/kg per dayFemale Sprague Dawley (SD) ratsIncreased the level of GDNF in DA neurons, contributing to neuroprotection in the MPP+ rat model of PD, with activation of mammalian target of rapamycin complex 1 and pre-treatment with naringin could attenuate the level of TNF-α in the substantia nigra of MPP+-treated brains[75]Naringin80 mg/kgmale Wistar albino ratsNeuroprotective activity against rotenone-induced toxicity in the animals possibly through Nrf2-mediated pathway[76]Naringin50, 100 and 200 mg/kgSwiss albino miceReduction in haloperidol-induced cataleptic scores in both bar test and block test[77]Naringenin25, 50, 100 mg/kg/b.w, POmale C57BL/6J miceReversed the toxic effects of MPTP by reducing LPO levels and increasing the activities of glutathione reductase and catalase along with improved behavioral performance[78]Naringenin50 mg/kg, orallyalbino Wister ratsImproved oxidative stress status by decreasing MDA and increasing glutathione content[79]Naringenin50, 100 mg/kgmale Sprague-Dawley ratsBV-2 and MN9D cell linesInhibition of microglia-induced neuroinflammation via NLRP3 inflammasome inactivation[80]Naringenin25, 50, and 100 mMSH-SY5Y Cell LineReduction of the ROS production by decreasing oxidative stress markers such as LPO and NO and increasing SOD level. In addition, pretreatment with NGN decreased the inflammatory markers such as TNF-α and NF-κβ in MPP+-treated SH-SY5Y cells. Further, NGN decreased the pro-apoptotic marker—Bax—and increased the anti-apoptotic marker—Bcl-2—in MPP+-induced SH-SY5Y cells[81]Naringenin40 μMPrimary rat mesencephalic culturesDecreased TH-positive neurons and TUNEL positive neurons[82]Naringenin50 mg/kgMale Sprague-Dawley ratsRestoration of dopamine concentrations due to neuroprotective effects rather than compensatory effects by remaining TH-positive cells after 6-OHDA lesioning[83]Naringenin20, 40 and 80 mM (in vitro) 70 mg/kg, orally (in vivo)Human neuroblastoma SH-SY5Y cells and male C57BL/6 miceActivated Nrf2/ARE pathway in dopaminergic (in vitro)Up regulated protein levels of Nrf2/ARE genes (in vivo)Reduced striatal oxidative stress and subsequent apoptotic signalling cascades in striatum (in vivo)[84]Naringenin12.5 μM and 25 μMSH-SY5Y Human Neuroblastoma cell lineDownregulation of the expression of some Parkinsonian genes such as casp9, lrrk2, and polg and upregulate pink1[85]Naringenin25, 50, and 100 mg/ kg/p.oMale C57BL/6J miceReduced NO content and restored SOD activity, also downregulated TNF-α and IL-1β expression[86]Naringenin10 and 50 µMFemale Wistar ratsEnhanced astroglial neurotrophic effects on DA neurons through the regulation of Nrf2 activation,[87]Anxiety and depressionNaringenin5, 10 and 20 mg/kgMale ICR miceIncreased hippocampal 5-HT, NE and GR levels, and reduced serum corticosterone levels[88]Naringenin5, 10 and 20 mg/kgMale ICR miceUp-regulation of BDNF[89]Naringenin10, 25 and 50 mg/kgSwiss miceNaringenin (25–50 mg/ kg) ameliorated the hypolocomotion, depressive- and anxiety-like behaviors in hypoxic miceNaringenin (10 mg/kg) increases BDNF expression but did not significantly (*p* < 0.05) alter corticosterone and catalase contents. The increased expressions of iNOS and NF-kB as well as loss of amygdala neuronal cells were reduced by naringenin (10 mg/kg)[90]Naringin10 mg/kgAdult male Swiss miceAlleviation of the depressive and anxiogenic behaviors evidenced by the increased preference to sucrose and open arm entries and duration in SPT and EPM respectively[91]Naringin25–100 mg/kg, i.pSwiss miceIncreased the levels of GAD67, glutathione and decrease AChE activities, pro-inflammatory cytokines (TNF-α, IL-6), malondialdehyde, nitrite concentrations[92]Naringenin50 mg/kg/dayAdult male Wistar ratsMitigation of morphological anomalies in the hippocampal CA1 region and cortex and upregulation of BDNF, Shh, GLI1, NKX2.2, and PAX6[93]Huntington’s diseaseNaringenin0.2, 0.4 mMC3H10T1/2 cellsSuppression of the protein aggregation caused by EGFP-polyQ97 in mammalian cells.[94]Naringin50, 100 mg/kgMale Wistar ratsProtection against 3-nitropropionic acid induced neurotoxicity via nitric oxide mechanism[95]Naringenin50 mg/kg b.w, POAlbino Wistar ratsImprovement of the behavioral function and restored the activity of MAO and 5-HT levels and reduction of the activation of astrocytes against 3-NP induced neurotoxicity[96]Naringin40, and 80 mg/kgAdult male Sprague-Dawley ratsModulation of oxido-nitrosative stress, neuroinflammatory, apoptotic markers and mitochondrial complex activity[97]Naringin(80 mg/kg b.w/day, orally)Male Wistar ratsEnhancement of phase II and antioxidant gene expressions via Nrf2 activation[98]Naringin10 µMPC12 cellsModulation in expressions of B-cell lymphoma 2 and Bcl-2-associated X protein and enhancement of the nuclear translocation of Nrf2[99]Ischemic brain injuryNaringin50 and 100 mg/kgMale Wistar ratsRestoration of reduced glutathione and catalase activity and mitochondrial enzyme activities in cortex, striatum, cerebellum[100]Naringin106 mg/kg/daymale C57BL/6 strain miceSuppression of neuronal cell death, reversed the reduction in the level of phosphorylated calcium-calmodulin-dependent protein kinase II, had the tendency to reverse the reduction in the level of glutathione, and blockade of excessive activation of microglia and astrocytes[101]Naringin40, 80 mg/kgMale Sprague-Dawley ratsImprovement of early brain injury (EBI), including subarachnoid hemorrhage (SAH) severity, neurologic deficits, brain edema and blood-brain barrier integrity by attenuating SAH-induced oxidative stress and apoptosis, and reduction of the oxidative damage and apoptosis by inhibiting the activation of MAPK signaling pathway[102]Naringenin50 and 100 mg/kgMale Sprague–Dawley ratsDown-regulation of NOD2, RIP2, NF-κB, MMP-9 and up-regulation of claudin-5 expression[103]Naringin80, 120, or 160 mg/kg/ daySH-SY5Y cellsReduced 3-nitrotyrosine formation, NADPH oxidase, and iNOS expression. Increased nNOS, p47, and p67 expression. Decreased mitophagy[104]Naringin100 mg/kg/dayAdult Wistar male ratsContinual treatment increased SOD activity, decreased MDA, NO, iNOS, and IL-1β. It also improved rats’ behavioral performance[105]Spinalcord injuryNaringin20, 40 mg/kgFemale Sprague-Dawley ratUpregulation of the expression of NKx2.2 and 2′3′-cyclic nucleotide 3′-phosphodiesterase, and inhibition of β-catenin expression and GSK-3β phosphorylation[106]Naringenin5, 10, 15 mMMale Wistar ratsSuppression of MMP-9 activity and upregulation of GSH, catalase and MMP-2 activation[107]Naringenin50–100 mg/kgFemale Wistar ratsRepression of miR-223[108]Naringin25, 50, and 100 mg/kgAdult Sprague Dawley ratsReduction of TNF-α, IL-8 as well as MDAcontent and elevation of IL-10 as well as SOD activity[109]Chronic hyperglycemic peripheral neuropathyNaringenin50, 100 and 200 mg/kgMale Sprague Dawley ratsInhibition of upregulated expression of TNF-α, IL-1β and MCP-1 level; GFAP and Mac-1 mRNA expression[110]Naringenin25 and 50 mg/kgMale Sprague Dawley ratsIncrease GSH level and decrease MDA and NO level[111]Naringin50 and 100 mg/kg, b.wRat model of OXL-induced peripheralneuropathyImproved the level of superoxide dismutase, catalase, glutathione peroxidase, nuclear factor erythroid 2-related factor 2, Heme oxygenase-1, nuclear factor-κ B, tumor necrosis factor-α, interleukin-1β, Bax, Bcl-2, caspase-3, paraoxonase, mitogen-activated protein kinase 14, neuronal nitric oxide synthase (nNOS), acetylcholinesterase, and arginase 2[112]4-HNE: 4-hydroxynonenal, TBARS: Thiobarbituric reactive substances, H_2_O_2_: Hydrogen peroxide, PC: Protein carbonyl, GSH: Reduced glutathione, GPx: Glutathione peroxidase, GR: Glutathione reductase, GST: Glutathione-S-transferase, SOD: Superoxide dismutase, CAT: catalase, GSK-3β: Glycogen synthase kinase-3β, ROS: Reactive oxygen species, AchE: Acetylcholine esterase, DA: Dopaminergic, LPO: Lipid peroxidation, MPTP: 1-methyl-4-phenyl-1,2,3,6-tetrahydropyridine, 5-HT: Serotonin, NE: Norepinephrine, BDNF: brain-derived neurotrophic factor, MDA: Malondialdehyde.


### 5.3. Cerebral Ischemia

Cerebral ischemia is a disorder that may trigger a cascade of unfavorable biochemical responses in the brain, leading to malfunction of key brain regions and, commonly, neuropathy [113,114]. An inflammatory reaction and the production of ROS following an ischemia event may damage brain tissue and lead to neuronal death [115]. Ischemia-induced damage involves several kinases, including mitogen-activated protein kinases, extracellular signal-regulated kinases, signal transducers and activators of transcription 1, calcium/calmodulin-dependent kinases, etc. ROS initiate the caspase cascade and encourage the synthesis of pro-inflammatory cytokines including interleukin (IL)-1, IL-6, and tumor necrosis factor-alpha, all of which contribute to cell death. Although progress has been made, a full understanding of the molecular mechanisms behind post-ischemic neuronal damage remains elusive. However, naringin and naringenin have been shown to have a neuroprotective effect after ischemia [116,117,118,119]. 

Naringin lowers cholesterol, prevents blood clots, and improves blood circulation and nutrient supply [100]. In addition, naringin protects against central nervous and cardiovascular diseases. To investigate the role of NFKB1 in OGD/R + injured PC12 cells, the previous study measured the components of the HIF-1α/AKT/mTOR signal path. HIF-1α, phosphorylated AKT, and mTOR were all higher in the OGD/R + naringin, OGD/R + si-NFKB1, and OGD/R + naringin + si-NFKB1 groups than in the OGD/R group. Significantly higher levels of HIF-1α, activated kinase AKT, and mTOR signals were expressed in the OGD/R + si-NFKB1 group (*p* < 0.01). In conclusion, naringin targets NFKB1 and modifies PC12 cell proliferation and apoptosis by affecting HIF-1α, p-AKT, and p-mTOR levels [120,121,122,123].

There is evidence that naringenin may help reduce the adverse effects of oxidative stress on the body, making it a potentially beneficial treatment option for various chronic illnesses. In addition to modulating the activity of antioxidant enzymes and regulating the expression of antioxidant genes, the flavonoid may have direct antioxidant effects, such as scavenging reactive species and reducing oxidative damage [124,125]. When Nrf2 is activated, antioxidant defenses are enhanced by maintaining redox equilibrium and decreasing oxidative stress and inflammation. TNF-α and IL-1β, two key inflammatory activators, are elevated in response to oxidative stress, whereas naringenin suppresses mRNA expression in the substantia nigra, hippocampus, and BV2 microglial cells [126,127]. For example, studies on rat hippocampus and BV cells have revealed that naringenin reduces NF-kB activation. Naringenin inhibited NF-kB, COX-2, iNOS, and their immunoreactivity, preventing the inflammatory cascade that leads to neurological diseases [51,128].

### 5.4. Anxiety and Depression

Anxiety and depression are two of the most common mental illnesses, both of which have complex origins at the intersection of several biological systems. Li et al. performed the first studies on the antidepressant effects of naringin and naringenin. The chemicals were tested on mice models of depression brought on by chronic unpredictable mild stress (CUMS) [129]. Anxiety may cause a variety of uncomfortable physical and emotional symptoms, including but not limited to: irritation, impatience, weariness, difficulty concentrating, a racing heart, chest pain, and an upset stomach. Anxiety comes in a variety of forms, and each is treated differently [130]. The serotonergic and noradrenergic systems have been connected to mood disorders such as depression and anxiety. The serotonergic system has far-reaching effects on cognitive processes in the brain, in addition to its role in regulating mood and appetite. Memory and focus are only two of the cognitive functions that are controlled by the noradrenergic system. Increases in serotonin (5-HT) and norepinephrine (NE) receptors, activation of brain-derived neurotrophic factor (BDNF), and decreased blood corticosterone are hypothesized to underlie NRG’s antidepressant-like effects [131,132]. It inhibits monoamine oxidase, which may also help those who are depressed. Increased rearing activity, decreased immobility, and increased social communication was seen in mice administered NRG intraperitoneally (at doses of 2.5, 5, and 10 mg/kg), which is consistent with anti-depressant-like and anxiolytic-like effects. Reductions were seen in nitrosative stress, lipid peroxidation, and cholinergic transmission. Mental diseases, often known as mental illnesses or psychiatric disorders, are characterized by persistent patterns of thinking or behavior that significantly impair an individual’s capacity to function in everyday life. Both the frequency and length of time that these symptoms will persist are unknown at this time. Various diseases and disorders have been identified, and each has its signs. In some instances, seeking the assistance of a clinical psychologist or psychiatrist specializing in evaluating and managing mental health conditions may be beneficial [133,134,135,136].

Neuronal inflammatory mediator release exacerbates tissue damage and reactive oxygen and nitrogen species (ROS/RNS) production, perpetuating the neuronal degeneration in stress-induced neuropsychiatric diseases, including depression and cognitive loss. Decreased antioxidant defenses in neurons promote neuroinflammation and neurodegeneration [137]. Several mediators and intracellular signaling molecules have been connected to neuroinflammatory responses to hypoxia damage. When hypoxia occurs, inflammatory transcription factors, including the NF-κB pathway, are activated, increasing pro-inflammatory cytokine production [137]. Physical changes in the brain’s dendritic arbourization and synaptic architecture have been linked to psychological and neurological issues, including depression, anxiety, and memory loss. Naringenin, a dietary flavanone, may be abundant in various foods, including citrus fruits, vegetables, and other berries and nuts [138]. Chronic illnesses and ailments may benefit from consuming a diet high in NG-rich fruits and vegetables. Animal pharmacokinetic studies have shown that NG rapidly undergoes intermediate glucuronide metabolism in the liver and readily crosses the blood–brain barrier (BBB). NG’s high permeability across the BBB has been attributed to its association with a broad range of CNS effects. However, the oral bioavailability of naringenin is limited by its metabolism in the liver and its degradation by bacterial enzymes in the colon. Lowered levels of inflammatory mediators were seen in rats, including TNF-α, cyclooxygenase-2, and inducible nitric oxide synthase (iNOS). Studies of naringenin’s effects on the brain and spinal cord show it may help treat various neurological disorders [139,140,141].

### 5.5. Schizophrenia

Schizophrenia is a disabling brain illness characterized by a wide range of neurotic symptoms including but not limited to hallucinations, delusions, cognitive impairment, disorganized speech, and aberrant motor activities [142]. Many different causes contribute to the development of schizophrenia. These include structural brain abnormalities, impaired neurotransmission, and stress-induced signaling cascades. Disruptions in epidermal growth factor (EGF) signaling and abnormalities in the processing or expression of the EGF receptors ErbB1 and ErbB2 are common in all schizophrenias [143]. As the pathophysiology of schizophrenia worsens, oxidative stress has been suspected to be a contributing factor. Several signs of oxidative stress, including ROS, reduced antioxidant enzyme activity (catalase), depleted glutathione, and oxidized lipids, have been associated with schizophrenia [144].

Schizophrenia and apoptosis have been connected via both intrinsic (mitochondrial death) and extrinsic (death receptor) pathways [145]. Cytochrome C interacts with pro- apoptotic and anti-apoptotic proteins to trigger the release of activated caspase-3 (primarily Bax and Bcl-2). Diabetes was prevented in streptozotocin-treated rats by administration of naringin, which inhibited the production of inflammatory and oxidative stress mediators [146]. Reduced free radical production, decreased release of proinflammatory cytokines (such as interleukin-6 and TNF-α), and down-regulation of inflammatory proteins such as NF-κB have all been linked to its anti-inflammatory effects in diabetic, chronic bronchitis, and walker carcinosarcoma rats. To evaluate if naringenin protects interendothelial tight junctions, we analyzed the expression and localization of ZO-1, occludin, claudin-1, and claudin-2 across experimental groups. ZO-1 protein expression was significantly reduced in the TNF-α treated group compared to the control group (*p* < 0.05) [147,148,149,150,151].

However, in TNF-α induced RIMVECs (*p* < 0.05), treatment with naringin dramatically increased ZO-1 protein expression. Immunofluorescence’s structured cell death also showed ZO-1 distribution. As proposed here, the medication that acts as a positive allosteric modulator of GABA neurotransmission from chandelier neurons is thought to improve the function of dorsolateral prefrontal cortex circuitry in people with schizophrenia by increasing gamma-band synchronization of pyramidal neuron activity [152,153,154,155].

## 6. Concluding Remarks and Future Directions 

Although technological advances have substantially sped up research on phytochemicals, we still have a long way to go before we gather more definitive evidence regarding the neurotherapeutic benefits of herbal medicines. Our data and other researchers’ data lead us to believe that naringin and naringenin may be beneficial as neurotherapeutic medications because of their ability to alter several signaling pathways. The outcomes thus far are in line with this theory.

Despite the limitations of ongoing clinical studies, naringenin and naringin are promising therapies for various neurological conditions, including AD, PD, cerebral ischemia, anxiety, depression, schizophrenia, and chronic hyperglycemic peripheral neuropathy. Given these obstacles, it is essential that pharmacokinetic research on naringin and naringenin administration be performed, that more accurate dosage designs for different illnesses be developed, and that innovative drug delivery strategies be developed to boost bioavailability in healthcare situations.

## Figures and Tables

**Figure 1 life-13-00099-f001:**
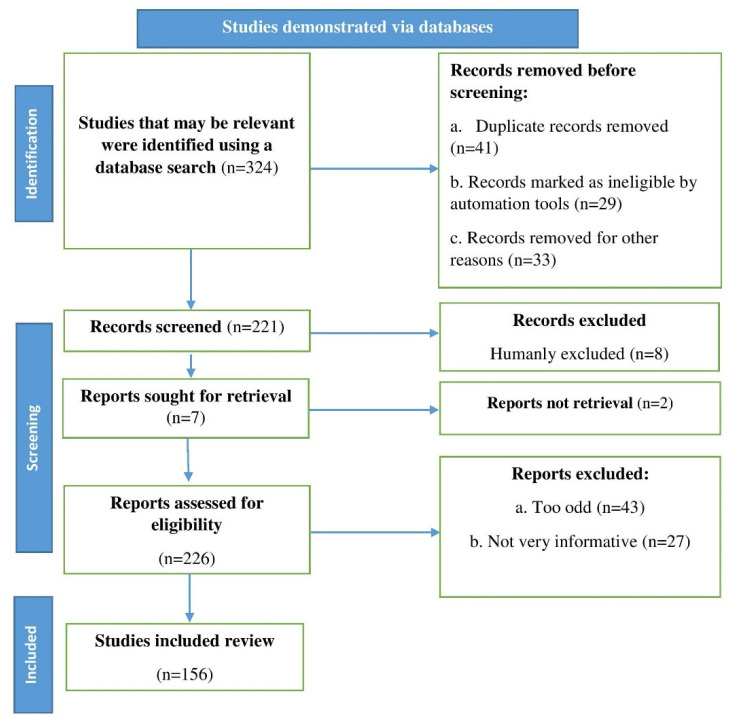
Flowchart depicting the steps to be followed when selecting published data for the current study; n stands for the number of literature reports.

**Figure 2 life-13-00099-f002:**
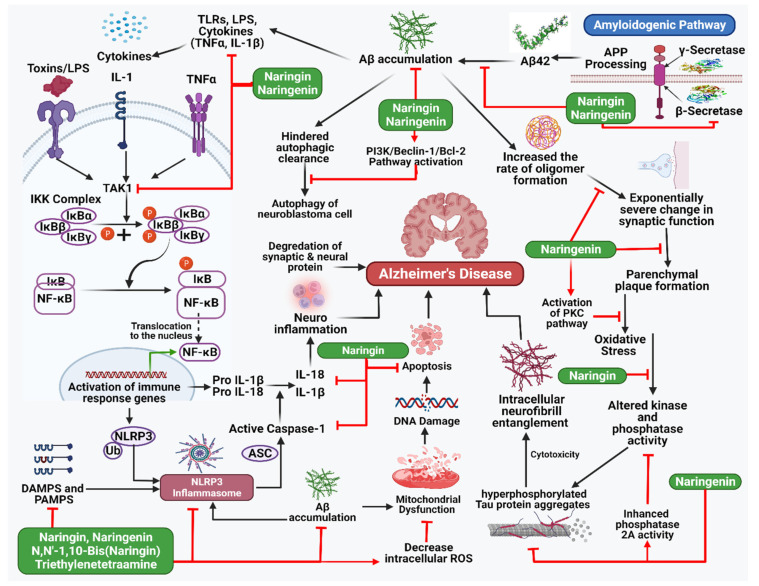
Illustration representing the site of action of naringin and naringenin in Alzheimer’s disease pathway.

**Figure 3 life-13-00099-f003:**
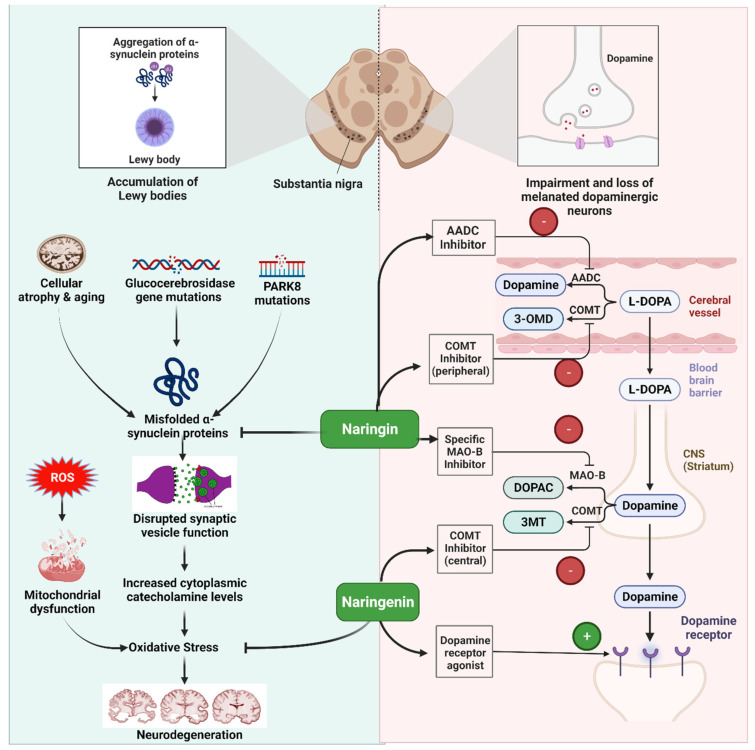
Illustration representing the site of action of naringin and naringenin in Parkinson’s disease.

## Data Availability

All data used to support the findings of this study are included within the article.

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
