# Peer review of "Naringin and Naringenin Polyphenols in Neurological Diseases: Understandings from a Therapeutic Viewpoint"

_life, 2022, doi:10.3390/life13010099_

Round 1

Reviewer 1 Report

In this review, the authors discuss the potential applications of naringin and naringenin in a broad spectrum of neurological disorders. In general, this review suffers from imprecise use of terminology and its tone is not always most appropriate, being at times too colloquial, and it is, in general, uneven. The authors' point is, in some cases, conveyed in an unclear manner, making it difficult to understand the relevance of reported results.  

In detail:

page 2, 59-64: this paragraph is counterintuitive. the authors should deepen their explanation of why "adaptation to injuries in the peripheral nervous system may be feasible due to neurodegeneration. The authors should also immediately point out the potential utility of polphenols and flavonoids: as it stands the paragraph feel disjointed, and the point the authors want to make is not entirely clear.
Page 2, 80-84: the authors should stress the connection between the previous paragraph and this one. the "your" in l.83 should be removed.
Page 4, 181-182: it would be more precise to say that memantine is an "antagonist of the NMDA glutamate receptor", not simply "an anti-glutamate receptor". IMPORTANT: AChE stands for acetylcholinesterase, not acetylcholesterol!! Finally, aricept is a commercial name; the authors should report the name of the active compound - donepezil.
Page 5, 199: what is kinetic acid? kainic acid is used in animal models of epilepsy.
Page 5, 202: the authors repeat where naringenin is found. If there is a lack of data regarding this flavonoid, the authors should state so.
Page 5, 208-233: this paragraph is completely disjointed from the rest of the review. Are the authors talking about naringenin or norepinephrine? It is unclear what this sudden focus on norepinephrine is about.
Page 6, 241-242: it is imprecise to say that theories regarding oxidative stress and mitochondrial malfunction have been "debunked"; rather, they are now understood as being downstream of multiple genetic risk factors. This claim also undermines the authors' interest in natural antioxidants such as polyphenols - why should they be of use if the role of oxidative stress in PD pathogenesis has been "debunked"?
Page 16, 302: the meaning of the sentence "it is now possible for humans to use narcissus" is unclear.
Page 16, 303-310: what is the meaning of "to investigate the role... we measured the components"?? This is not the author's research work!
Page 18, 410-436: the paragraph regarding ROS generation, neuroinflammation and MMPs adds nothing to the discussion and should either be moved and rephrased, or deleted entirely.

In general, the authors should avoid the use of contractions ("it's" etc.) and colloquial expressions ("zeroed in", "canceled out") and a rewrite of several parts of the manuscript is required. References and bibliography should be recompiled following MDPI guidelines. The necessary revisions are extensive, and combined with the lack of perspectives that the conclusions of this review present, make it very hard to recommend this paper for publication on life, even with major revisions.

Author Response

In this review, the authors discuss the potential applications of naringin and naringenin in a broad spectrum of neurological disorders. In general, this review suffers from imprecise use of terminology and its tone is not always most appropriate, being at times too colloquial, and it is, in general, uneven. The authors' point is, in some cases, conveyed in an unclear manner, making it difficult to understand the relevance of reported results.

In detail:

page 2, 59-64: this paragraph is counterintuitive. the authors should deepen their explanation of why "adaptation to injuries in the peripheral nervous system may be feasible due to neurodegeneration. The authors should also immediately point out the potential utility of polphenols and flavonoids: as it stands the paragraph feel disjointed, and the point the authors want to make is not entirely clear.

Reply: Thanks for pointing it out; we have made edits to cover it. We have also made changes to ensure consistency allowing better readability (changes highlighted in the revised manuscript).

Page 2, 80-84: the authors should stress the connection between the previous paragraph and this one. the "your" in l.83 should be removed.

Reply: Thanks for the detailed suggestions. We agree with the respected reviewer and removed the mentioned section.

Page 4, 181-182: it would be more precise to say that memantine is an "antagonist of the NMDA glutamate receptor", not simply "an anti-glutamate receptor". IMPORTANT: AChE stands for acetylcholinesterase, not acetylcholesterol!! Finally, aricept is a commercial name; the authors should report the name of the active compound - donepezil.

Reply: Thanks for the suggestions. We agree with the respected reviewer and revised it as per suggestions.

Page 5, 199: what is kinetic acid? kainic acid is used in animal models of epilepsy.

Reply: Thank you for pointing it out; we have made the correction as per the suggestion.

Page 5, 202: the authors repeat where naringenin is found. If there is a lack of data regarding this flavonoid, the authors should state so.

Reply: Thank you for pointing it out; we have made the correction as per the suggestion.

Page 5, 208-233: this paragraph is completely disjointed from the rest of the review. Are the authors talking about naringenin or norepinephrine? It is unclear what this sudden focus on norepinephrine is about.

Reply: Thank you for pointing it out; we have made the correction as per the suggestion.

Page 6, 241-242: it is imprecise to say that theories regarding oxidative stress and mitochondrial malfunction have been "debunked"; rather, they are now understood as being downstream of multiple genetic risk factors. This claim also undermines the authors' interest in natural antioxidants such as polyphenols - why should they be of use if the role of oxidative stress in PD pathogenesis has been "debunked"?

Reply: Thanks for the detailed suggestions. We agree with the respected reviewer. We have made changes to hone the content and make it crisp. We have also edited grammar to ensure better readability. We are hopeful that the reviewer and the editor(s) will like it better now. We are confident that the readership of the journal will relish such a detailed table having a stratified summary of the literature on the topic in a single place.

Page 16, 302: the meaning of the sentence "it is now possible for humans to use narcissus" is unclear.

Reply: Thanks for the suggestions. We have edited the grammar to ensure better readability.

Page 16, 303-310: what is the meaning of "to investigate the role... we measured the components"?? This is not the author's research work!

Reply: Thanks for the suggestions. We have edited it as per suggestions.

Page 18, 410-436: the paragraph regarding ROS generation, neuroinflammation and MMPs adds nothing to the discussion and should either be moved and rephrased, or deleted entirely.

Reply: Thanks for the detailed suggestions. We agree with the respected reviewer. We have revised as per suggestions.

In general, the authors should avoid the use of contractions ("it's" etc.) and colloquial expressions ("zeroed in", "canceled out") and a rewrite of several parts of the manuscript is required. References and bibliography should be recompiled following MDPI guidelines. The necessary revisions are extensive, and combined with the lack of perspectives that the conclusions of this review present, make it very hard to recommend this paper for publication on life, even with major revisions.

Reply: We are glad that the respected reviewer liked our work, and we express our thanks for their valuable suggestions to help improve our manuscript. We again express our gratitude to the reviewers and the editor for the detailed help in improving the work. We look forward to hearing from you regarding our submission and responding to any further questions and comments you may have.

Reviewer 2 Report

The article titled: Naringin and Naringenin Polyphenols in Neurological Diseases: Understandings from Therapeutic View point presents valuable information. However, the Authors should improve some parts and rearrange them. Below I give my comments.

1. Abstract is well written. However, I suggest putting in evidence the aim of the prepared review at the end of this part.

2. The authors should add the Material and methods Paragraph. At this point, they should describe the databases used to prepare the manuscript, the keywords, and the other information which defines well how the Authors collected the necessary scientific material.

3. The title of paragraph 2, "Naringin and Naringenin Derivatives," suggests that naringin, naringenin, and other derivatives compounds will be described. However, only these two compounds are presented. If the content of paragraph 2 is without changes, the Authors should correct the title.

4. Section 3. Only the first part is connected with the primary goal of this Paragraph – the botanical sources of naringin and naringenin. The presentation of sources of derivatives of these flavonoids is also justified. Why do Authors describe norepinephrine? And why they write about the biological activity of this substance in the Paragraph titled: Botanical Sources. Norepinephrine is not a flavonoid at all. In my opinion, paragraph 3 should be better organized and rewritten.

5. The section "4.6. Chronic Hyperglycemic Peripheral Neuropathy" sounded very interesting, but after reading it, I have the impression that any important information about naringin and naringenin was not presented…

6. Table 1. It summarizes the interesting results on naringenin or naringin AD or PD activity. However, sometimes the arrangement is disrupted – in the same rubric, naringin and naringenin activity are shown alternately.  

7. The more recent references should be sometimes presented, e.g., "Alzheimer's disease (AD) is the most common type of dementia in the industrialized world and is a severe public health problem (Thrall 2005)". In this part and the others similar, Authors should incorporate the new citations (no older than 5-10 years from this moment).

Author Response

The article titled: Naringin and Naringenin Polyphenols in Neurological Diseases: Understandings from Therapeutic View point presents valuable information. However, the Authors should improve some parts and rearrange them. Below I give my comments.

  1. Abstract is well written. However, I suggest putting in evidence the aim of the prepared review at the end of this part.

Reply: We express our thanks to the reviewer for his detailed insights, and we made the changes accordingly.

  1. The authors should add the Material and methods Paragraph. At this point, they should describe the databases used to prepare the manuscript, the keywords, and the other information which defines well how the Authors collected the necessary scientific material.

Reply: We express our thanks to the reviewer for his detailed insights, and we made the changes accordingly.

  1. The title of paragraph 2, "Naringin and Naringenin Derivatives," suggests that naringin, naringenin, and other derivatives compounds will be described. However, only these two compounds are presented. If the content of paragraph 2 is without changes, the Authors should correct the title.

Reply: Thank you for pointing it out; we have made the correction as per the suggestion.

  1. Section 3. Only the first part is connected with the primary goal of this Paragraph – the botanical sources of naringin and naringenin. The presentation of sources of derivatives of these flavonoids is also justified. Why do Authors describe norepinephrine? And why they write about the biological activity of this substance in the Paragraph titled: Botanical Sources. Norepinephrine is not a flavonoid at all. In my opinion, paragraph 3 should be better organized and rewritten.

Reply: Thanks for the detailed suggestions. We agree with the respected reviewer. We have made changes to hone the content and make it crisp.

  1. The section "4.6. Chronic Hyperglycemic Peripheral Neuropathy" sounded very interesting, but after reading it, I have the impression that any important information about naringin and naringenin was not presented…

Reply: Thanks for the detailed suggestions. We agree with the respected reviewer. We have made changes to hone the content and make it crisp.

  1. Table 1. It summarizes the interesting results on naringenin or naringin AD or PD activity. However, sometimes the arrangement is disrupted – in the same rubric, naringin and naringenin activity are shown alternately.  

Reply: Thanks for pointing it out; we have made edits to cover it, and Table 1 used has now been expanded as per suggestions.

  1. The more recent references should be sometimes presented, e.g., "Alzheimer's disease (AD) is the most common type of dementia in the industrialized world and is a severe public health problem (Thrall 2005)". In this part and the others similar, Authors should incorporate the new citations (no older than 5-10 years from this moment).

Reply: Thanks for pointing it out; we have made edits and updated the references.

We again express our gratitude to the reviewers and the editor for the detailed help in improving the work. We look forward to hearing from you regarding our submission and responding to any further questions and comments you may have.

Reviewer 3 Report

The authors make a decent attempt on reviewing Naringins  and their phytochemical compounds, viz Glycosides and review their role in various diseases/functions

The authors give  a gist of progress in lieu of treatment, and focus on neurodegenerative diseases as potent targets.  The review is written well with two excellent Figures. 

However, keeping in viw of the figures, a tabular description is warranted 

Scores on a scale of 0-5 with 5 being the best 

Language: 3

Novelty: 4

Scope and relevance: 4

brevity: 4

Attached are edits, the manuscript needs to be checked for language.

Author Response

viz Glycosides and review their role in various diseases/functions

The authors give  a gist of progress in lieu of treatment, and focus on neurodegenerative diseases as potent targets.  The review is written well with two excellent Figures. 

However, keeping in viw of the figures, a tabular description is warranted 

Scores on a scale of 0-5 with 5 being the best 

Language: 3

Novelty: 4

Scope and relevance: 4

brevity: 4

Attached are edits, the manuscript needs to be checked for language.

Reply: Thanks for the detailed suggestions. We agree with the respected reviewer. We have made changes to hone the content and make it crisp. We have also edited grammar to ensure better readability. We are hopeful that the reviewer and the editor(s) will like it better now. We are confident that the readership of the journal will relish such detailed content having a stratified summary of the literature on the topic in a single place.

We again express our gratitude to the reviewers and the editor for the detailed help in improving the work. We look forward to hearing from you regarding our submission and responding to any further questions and comments you may have.

Reviewer 4 Report

The authors based their review on the premise that two flavonoids: naringin and naringenin, are associated with various biological functions, such as neuroprotective, antioxidant, anticancer, antiviral, antibacterial, anti-inflammatory, antiadipogenic, and cardioprotective. They propose that these natural compounds may constitute a promising strategy for treating neurological conditions, including neurodegenerative diseases (e.g., Alzheimer's and Parkinson's) and other neurological conditions (e.g., anxiety, depression, and schizophrenia, among others). This review highlights the role of naringin and naringenin as neurotherapeutic agents. A comprehensive analysis of advances in understanding the molecular mechanisms associated with the protective benefits of these flavonoids in different neurological conditions is also described.

Given the above, publication of the current version of the manuscript in "Life" is endorsed.

Author Response

The authors based their review on the premise that two flavonoids: naringin and naringenin, are associated with various biological functions, such as neuroprotective, antioxidant, anticancer, antiviral, antibacterial, anti-inflammatory, antiadipogenic, and cardioprotective. They propose that these natural compounds may constitute a promising strategy for treating neurological conditions, including neurodegenerative diseases (e.g., Alzheimer's and Parkinson's) and other neurological conditions (e.g., anxiety, depression, and schizophrenia, among others). This review highlights the role of naringin and naringenin as neurotherapeutic agents. A comprehensive analysis of advances in understanding the molecular mechanisms associated with the protective benefits of these flavonoids in different neurological conditions is also described.

Given the above, publication of the current version of the manuscript in "Life" is endorsed.

Reply: Thanks for the detailed suggestions. We agree with the respected reviewer. We have made changes to hone the content and make it crisp. We have also edited grammar to ensure better readability. We are hopeful that the reviewer and the editor(s) will like it better now. We are confident that the readership of the journal will relish such detailed content having a stratified summary of the literature on the topic in a single place.

We again express our gratitude to the reviewers and the editor for the detailed help in improving the work. We look forward to hearing from you regarding our submission and responding to any further questions and comments you may have.

Reviewer 5 Report

The review summarized the effect and mechanisms of naringin and naringenin polyphenols in neurological disease. It is interesting. Here are some issues need to be fixed.

1. The review mainly discussed Alzheimer and Parkinson diseases (from table), what about others, such as the brain inflammation, the hippocampus inflammation?

2. Section 2. The title is naringin and naringenin derivatives. I assume the authors will talk about their structures, their metabolism and absorption. But the authors talk their effect. It is not consistent with the title. If authors discuss therapeutic or preclinical studies, the structure, metabolism and absorption needed to be reviewed and discussed. So, this section should be changed or there should be a more section to talk about the absorption and metabolism of these compounds.

3. In table 1, there are many mistakes. authors should double check it carefully. 

      a. All this treatment is naringin and naringenin pure compound, or they are plant extract? need to be clarified since the ative compound concentration will be totally different.

b. what is p.o. and PO? Need to be consistent.

c.  dose is mg/Kg, it is to body weight or to diet?

d. for the reference (Yang et al. 2018), the dose column and mice model column are opposite.

e. there are so many abbreviations, some have full name, some not, the authors need to have footnotes to explain all of these abbreviations.

f. for reference (Heo et al. 2004a), it shows ROS and LDH, why later in reference (Guo and Sum 2021), the authors write full name of ROS and abbreviate it.

g. what is (Rahigude et al. 2012)s?

h. what is um/mL, if it is concentration, should be uM. Also, ml, mL, uM, mM all appeared in this table.

i, the row (Olsen et al, 2008), IC50?

j, the results summarized in Table 1 somehow looks messy, please re-ordered them by reference year.

k. bt.w and body weight both appeared.

4. If there are any human studies or epidemiological studies? If yes, authors need to include them.

5. If there are any biomarkers evaluating naringin uptake?

Author Response

The review summarized the effect and mechanisms of naringin and naringenin polyphenols in neurological disease. It is interesting. Here are some issues need to be fixed.

  1. The review mainly discussed Alzheimer and Parkinson diseases (from table), what about others, such as the brain inflammation, the hippocampus inflammation?

Reply: Thanks for pointing it out; we have made edits to cover it (changes highlighted in the revised manuscript).

  1. Section 2. The title is naringin and naringenin derivatives. I assume the authors will talk about their structures, their metabolism and absorption. But the authors talk their effect. It is not consistent with the title. If authors discuss therapeutic or preclinical studies, the structure, metabolism and absorption needed to be reviewed and discussed. So, this section should be changed or there should be a more section to talk about the absorption and metabolism of these compounds.

Reply: Thanks for pointing it out; we have made edits to cover it (changes highlighted in the revised manuscript).

  1. In table 1, there are many mistakes. authors should double check it carefully. 
  2. All this treatment is naringin and naringenin pure compound, or they are plant extract? need to be clarified since the ative compound concentration will be totally different.
  3. what is p.o. and PO? Need to be consistent.
  4. dose is mg/Kg, it is to body weight or to diet?
  5. for the reference (Yang et al. 2018), the dose column and mice model column are opposite.
  6. there are so many abbreviations, some have full name, some not, the authors need to have footnotes to explain all of these abbreviations.
  7. for reference (Heo et al. 2004a), it shows ROS and LDH, why later in reference (Guo and Sum 2021), the authors write full name of ROS and abbreviate it.
  8. what is (Rahigude et al. 2012)s?
  9. what is um/mL, if it is concentration, should be uM. Also, ml, mL, uM, mM all appeared in this table.

i, the row (Olsen et al, 2008), IC50?

j, the results summarized in Table 1 somehow looks messy, please re-ordered them by reference year.

  1. bt.w and body weight both appeared.

Reply: Thank you for pointing out the error. We have proofread the manuscript in detail again, and to our surprise, we found many such errors which we skipped. We apologize for these errors; we have extensively revised the manuscript to correct them and we ensure you the current version is free of any spelling, grammar, or formatting errors (changes highlighted yellow in the revised manuscript).

  1. If there are any human studies or epidemiological studies? If yes, authors need to include them.

Reply: Thanks for the suggestions. We tried to include some as per suggestions.

  1. If there are any biomarkers evaluating naringin uptake?

Reply: Thank you for pointing out this. We did not find such.

We again express our gratitude to the reviewers and the editor for the detailed help in improving the work. We look forward to hearing from you regarding our submission and responding to any further questions and comments you may have.

Round 2

Reviewer 1 Report

The extensive changes made by the authors are satisfactory for publication, provided thorough proofreading is carried out.

Author Response

The extensive changes made by the authors are satisfactory for publication, provided thorough proofreading is carried out.

Reply: We are glad that the respected reviewer liked our work and we express our thanks for their valuable suggestions to help improve our manuscript. We again express our gratitude to the reviewers and the editor for the detailed help improving the work.

Reviewer 2 Report

Thank you to the authors for the amendments introduced to the manuscript. I find that in this form, the article can be published.

Author Response

Thank you to the authors for the amendments introduced to the manuscript. I find that in this form, the article can be published.

Reply: We are glad that the respected reviewer liked our work, and we express our thanks for their valuable suggestions to help improve our manuscript. We again express our gratitude to the reviewers and the editor for the detailed help in improving the work. Also, thank you very much for the acceptance.

Reviewer 5 Report

The author revised the manuscript.

For the table, it is better to write abbreviated names in the table to make it clear and simple but include all full names under the table as note. 

Author Response

The author revised the manuscript.

For the table, it is better to write abbreviated names in the table to make it clear and simple but include all full names under the table as note.

Reply: Thanks for pointing it out; we have made edits to cover it (changes highlighted in revised manuscript).

We again express our gratitude to the reviewers and the editor for the detailed help improving the work. We look forward to hearing from you in due time regarding our submission and to responding to any further questions and comments you may have.